# A Narrative Review of Screen Time and Wellbeing among Adolescents before and during the COVID-19 Pandemic: Implications for the Future

**DOI:** 10.3390/sports11020038

**Published:** 2023-02-06

**Authors:** Verity Y. Q. Lua, Terence B. K. Chua, Michael Y. H. Chia

**Affiliations:** 1School of Social Sciences, Singapore Management University, 10 Canning Rise, Singapore 179873, Singapore; 2Physical Education and Sports Science Academic Department, National Institute of Education, Nanyang Technological University, 1 Nanyang Walk, Singapore 637616, Singapore

**Keywords:** screen time, wellbeing, adolescents

## Abstract

The COVID-19 pandemic has disrupted the lives of many. In particular, during the height of the pandemic, many experienced lockdowns, which in turn increased screen time drastically. While the pandemic has been declared an endemic and most activities have been reinstated, there appears to still be elevated screen time among adolescents due to poor habits formed during the pandemic lockdowns. This paper explores the factors by which screen time affects well-being among adolescents and how the pandemic may have influenced some of these factors. For example, beyond having greater screen time, many adolescents have also reduced their physical activities and picked up poor sleeping habits. These findings highlight the importance of having integrated activity guidelines that go beyond limiting adolescents’ daily screen time. It is important to affirm that beyond ensuring that adolescents limit their daily screen time, they are also meeting physical activity and sleep guidelines so that they achieve a holistic sense of wellbeing.

## 1. Introduction

The COVID-19 pandemic has disrupted the lives of many; most drastically, many cities experienced lockdowns, curfews, and school closures. Whilst being indoors and social distancing measures have mitigated the spread of COVID-19, such measures have also caused drastic changes in lifestyles. In many cases, this has led to the cultivation of habits such as a greater engagement with digital devices. Many schools have also relied on technology to deliver lessons virtually. Furthermore, being unable to go outdoors, many have turned to digital media for entertainment and to stay socially connected [1]. It has been estimated that during the pandemic, total time spent on digital devices among adolescents (aged 11–17 years old) increased by 0.9 h per day [2].

Importantly, while lockdowns have mostly been lifted, there is little evidence to suggest that average screen time has returned to pre-pandemic levels [3]. Indeed, despite the easing of lockdown restrictions, global average screen time in the third quarter of 2021 was still higher than in the third quarter of 2020 by 1.0% [3]. While the debate about whether screen time is inherently harmful remains, much research has suggested that an increase in screen time is often positively correlated with other behaviours that are detrimental to holistic wellbeing [4,5]. According to the displacement hypothesis (e.g., [6]), time spent on digital media displaces time used to engage in other behaviours that promote wellbeing, such as physical activity and offline socialising. In line with this hypothesis and the trend that screen time has not recovered to pre-pandemic levels, global average step counts were also lower post-lockdown (i.e., in late 2021 and early 2022) relative to pre-pandemic step counts measured during mid-2019 [7]. This highlights that screen time habits developed during the pandemic may have lowered time spent engaging in physical activity, even after lockdown orders have largely been lifted.

These trends are particularly concerning for adolescents (10–19-year-olds), given that much research has shown that habits developed during these formative years are often long-lasting. The habits formed in adolescents concerning screen time, physical activity engagement, and dietary habits, for example, have been shown to continue into adulthood [8,9,10]. Indeed, the adolescent period has been deemed by many as part of one’s “impressionable years” [11], whereby adolescents are particularly susceptible to change. Indeed, major changes in lifestyles brought about by the pandemic are likely to have profound and large impacts on adolescents in particular. Hence, the cultivation of poor habits during the height of the pandemic can have potentially far-reaching effects on today’s adolescents.

Taken together, there is a pressing need to examine the existing literature concerning the (1) mechanisms by which screen time affects well-being among adolescents and (2) how the pandemic has influenced some of these mechanisms for a better understanding of how we might promote better holistic wellbeing among adolescents in today’s world. Indeed, while there have been some existing works examining the relationship between screen time and well-being among adolescents, these works are often presented in the form of individual studies examining specific samples. Yet, there is much potential in comparing and contrasting these studies, given that considering multiple studies and meta-analyses in conjunction with one another lends greater confidence in the conclusions we draw. Thus far, there have been limited works synthesising this body of literature and considering the existing evidence as a whole. Hence, the present work seeks to present a digestible narrative review of the current literature by synthesising studies examining the screen time trends among adolescents before and during the pandemic. The review consists of three main parts. The first concerns the *trends of screen time across pandemic stages*, which examines if and how screen time among adolescents has changed from pre-pandemic levels. The second concerns the *impact of screen time on adolescents’ wellbeing*, which synthesises the relationship between screen time and physical activity, sleep, and socioemotional outcomes among adolescents and examines if the pandemic has affected these associations. The third concerns the suggestions for *future directions* based on the findings of the present narrative review.

## 2. Methods

The present review was guided in part by the general framework for writing narrative reviews recommended by Ferrari (2015). Approximately 17,500 records were identified through Google Scholar based on the keywords used to extract relevant articles. In the first phase of screening, approximately 70 abstracts were screened for inclusion, and records were excluded if they were irrelevant (e.g., not examining screen time) or relatively lower powered (e.g., individual studies with Western samples smaller than 1000 participants; the sample size criteria were smaller for individual studies examining participants from other nations due to the relatively smaller body of work meeting this requirement). Then, at least 45 full-text papers were examined to ensure the quality and relevance of the papers. A final 34 papers, consisting of reviews of meta-analyses, meta-analyses, and individual large-scale studies, were included in the final review (Figure 1). The papers included were judged to be adequately powered and relevant to the research questions at hand.

### 2.1. General Eligibility Criteria

#### Target Age Group

The present review primarily focuses on adolescents (i.e., ages 10–18 years). Nonetheless, studies that do not examine the whole age range and studies that compile other close age groups (e.g., children and adolescents, young adults, and adolescents) were included if there were no better comprehensive studies that could replace those studies. Thus, as long as studies examined adolescents, they were eligible for inclusion in the current narrative review.

### 2.2. Literature Search Process

#### 2.2.1. Information Sources

To ensure that publications from a wide range of electronic databases of published studies were considered, articles were retrieved via Google Scholar. The search strategy was structured using search terms like “well-being”, “wellness”, “adolescents”, “children”, “youth”, “screen-time”, “media use”, “physical health”, “sleep”, “pandemic”, “COVID-19”, “meta-analysis”, and multiple combinations of these keywords. The language of the publications searched was mainly English.

#### 2.2.2. Time Periods of Studies

To examine screen time trends during the pre-pandemic stages, studies published between the years 2016 and 2018 were included. As the cut-off, 2016 was chosen, as it would encapsulate research done three years prior to the pandemic, and the current literature on screen time would be able to capture three years since the start of the pandemic (i.e., 2020 to 2022). Additionally, this time range was selected due to the rapid change in technology in recent years. Specifically, comparing screen time trends prior to 2016 with current screen time trends may not be as accurate and valid, given that technological advancements in recent years have been particularly quick. Whilst the publication year was restricted, there were no restrictions on the studies included within each systematic review included, given that such a restriction would greatly limit the availability of reviews we could tap on. Hence, for the first section of the review examining pre-pandemic screen time levels, only studies published between 2016 and 2018 were selected.

To examine changes in screen time trends during the pandemic, only studies that directly compared pre-pandemic and pandemic screen time were considered. This is because the varied measures capturing screen time may not be directly comparable across multiple studies. Hence, it would be most valid to rely on existing works that utilise a standard measure of screen time both pre-pandemic and during the pandemic for a more accurate comparison. Thus, only studies that had their own measures of screen time both pre-pandemic and during the pandemic were included in the narrative review for this section.

### 2.3. Study Selection

#### Focus on Systematic Reviews and Highly-Powered Studies

To ensure that the studies selected were highly powered, there was a deliberate decision to focus primarily on meta-analyses and systematic reviews. Given that these types of works draw on multiple large samples across multiple countries, it was determined that these reviews would be the most representative and comprehensive forms of evidence. Where there were no systematic reviews available, large-scale panel and longitudinal studies were considered. Additionally, in light of the extant literature highlighting the differential impacts of digital media on adolescents’ wellbeing across cultures [13], the present literature review adopts a culturally nuanced approach in synthesising the existing literature. Where the systematic reviews did not include works from specific continents (e.g., no Asian studies were included), individual studies conducted in these geographical regions are discussed as well. In sum, the present narrative review has a specific focus on synthesising evidence from systematic reviews and highly-powered studies (see Table 1).

## 3. Discussion

### 3.1. The Impact of Screen Time on Adolescents’ Wellbeing

The present work delineates wellbeing into three major components—physical activity, sleep, and socioemotional wellbeing. These three aspects have been often associated with screen time [20], and are thus the focus of the present narrative review. In line with the previous section, only studies published during or after 2016 were included to ensure that the findings were relevant and recent. Studies that pooled the relationship between screen time and wellbeing before and during the pandemic as a whole (without differentiating the two) were excluded for this review. Additionally, to examine the effects of screen time on adolescents’ wellbeing during the pandemic, only studies specifically limited to the context of the pandemic were included. Hence, studies included in this section were exclusively considering the relationship between screen time and wellbeing either prior to the pandemic or within the context of the pandemic, rather than both concurrently.

Table 1 presents some of the seminal studies that are discussed in the present review.

### 3.2. Trends of Screen Time across Pandemic Stages

The first section of this literature review examines screen time trends before and during the pandemic. To ensure studies included in the review are adequately powered, the present work places a greater focus on meta-analytic studies. Nonetheless, to ensure a culturally nuanced review, individual studies from less-represented countries are discussed as well. To understand the current state of how much screen time adolescents have, we start by discussing studies examining the average daily screen time and the proportion of adolescents meeting screen time guidelines pre-pandemic. Next, we examine studies that compare pre-pandemic and pandemic levels of screen time among adolescents.

### 3.3. Pre-Pandemic Screen Time among Adolescents

Even prior to the COVID-19 pandemic, statistics from multi-nation studies suggest that adolescents spend more than the World Health Organisation’s recommended limit of two hours of screen time per day [21,22]. Meta-analyses and nation-wide studies examining the prevalence of adolescents meeting the 24-h WHO guidelines corroborate this. The largest meta-analytic study examining the prevalence of children and adolescents meeting 24-h Movement Guidelines comprised 63 studies across 23 countries [23]. Although they did not examine the proportion of adolescents who failed to meet the screen time guideline specifically, they found that 19.21% of adolescents and children met none of the three 24-h movement guidelines. Another meta-analytic study examining the prevalence of excessive screen time (television watching) specifically among Brazilian adolescents found that over 70% of adolescents consumed over two hours of television daily [24]. Similarly, based on a sample of 3096 adolescents from the Longitudinal Study of Australian Children and 5615 Canadian students from the 2017 Ontario Student Drug Use and Health Survey, 74.0% of Australian adolescents and 66.1% of Canadian adolescents did not meet the screen time recommendation [25,26]. 

There were fewer studies examining the prevalence of meeting the screen time recommendation specifically among adolescents from non-Western countries. However, existing studies still show that a substantial proportion of adolescents from non-Western countries fail to meet the guideline. For example, in a large-scale study conducted in China in 2017, approximately 35% of adolescents did not fulfil the two-hour screen time guideline [27]. Whilst this proportion is relatively lower, it is evident that a sizable proportion of adolescents do not meet the recommended guidelines.

Taken together, data from multiple parts of the world suggest that a substantial proportion of adolescents fail to meet the recommended screen time limit of two hours per day. Of importance, however, were the limited studies that examined the prevalence of adolescents meeting screen time recommendations, especially in non-Western countries. Additionally, there were fewer studies comparing screen time across different countries. Given that screen time is often measured differently across studies (e.g., including/excluding time spent on mobile devices beyond time spent watching television), it is difficult to compare screen time data derived from different studies. Indeed, briefly examining the Supplemental Materials provided by the meta-analysis conducted by Tapia-Serrano and colleagues [23] (see [23] Supplementary Table S5), it is jarring that studies conducted within the same country and age group appear to differ substantially on their findings regarding the prevalence of meeting the screen time guideline. While multiple factors may account for this observation, it is also possible that the measurement of screen time in these studies makes them incomparable. Nonetheless, generally speaking, most studies appear to suggest that even prior to the COVID-19 pandemic, many adolescents spent more than two hours a day on screen-related activities.

### 3.4. Adolescents’ Screen Time during the Pandemic

In light of pandemic restrictions and lockdowns, a host of studies have begun to examine whether screen time among adolescents has increased. On a more global scale, a literature review noted that all 31 studies (with data from over 20 countries) included in their review examining changes in screen time found increases in adolescents’ screen time during the pandemic compared to pre-pandemic [28]. These increases ranged from 55 min per day to 2.9 h per day. Trott and colleagues’ [2] meta-analysis based on 89 studies also found that primary-aged children (6–10 years) reported the greatest increases in both leisure and total screen time during the pandemic relative to other age groups—on average, they had an increase of 1.4 h of total screen time daily and an hour increase in leisure screen time daily.

Supporting these meta-analytic findings, region- and country-wide studies show similar trends. For example, a study utilising data from the Adolescent Brain Cognitive Development (ABCD) Study (which tracks over 5000 American adolescents) found substantial increases in recreational screen time during the first year of the pandemic compared to before the pandemic [29]. In a study examining differences in screen time pre-lockdown and during lockdowns in North Africa (specifically, the city of Constantine in Algeria), it was noted that the proportion of adolescents who exceeded the daily screen time recommendation of two hours per day increased from 3.3% to 5.5% [30]. A separate study examining Indonesian adolescents’ behaviour found that approximately 60.5% had greater screen time during the COVID-19 pandemic lockdowns [31]. Indeed, echoing these quantitative findings, interviews conducted with 20 parents in Yogyakarta (Indonesia) also revealed that parents were highly concerned about their adolescent children spending substantially more time on screen-based devices during the pandemic [32]. 

Taken together, existing research suggests that the pandemic has indeed increased adolescents’ screen time. These findings are particularly evident in higher-income countries, given that much more work has been done to examine the screen time trends among adolescents from these countries. Nonetheless, smaller-scale studies examining less-represented regions (e.g., South Africa) do appear to follow those found in the higher-income countries. Specifically, it appears that the COVID-19 pandemic has indeed increased the prevalence of adolescents not meeting the recommended screen time guideline of two hours per day. 

### 3.5. The Impact of Screen Time on Adolescents’ Wellbeing

To determine whether such an increase in screen time warrants attention, it would be important to examine the outcomes and correlates of screen time and to examine if these factors have changed in the context of current pandemic. More specifically, technology has the ability to both harm and promote adolescents’ wellbeing. In particular, during the pandemic, communicative technology (e.g., video calls, instant messaging) enabled many to continue with their day-to-day activities. Nonetheless, the robust associations between screen time and negative outcomes in the existing literature pose an intriguing question—do the benefits of screen-based activities outweigh the harm?

#### 3.5.1. Screen Time and Physical Wellness

One of the most well-researched associations of screen time in youth is that of physical activity. In line with the displacement hypothesis, many have posited that time spent on screen-based devices replaces the time adolescents spend engaging in physical activity. Indeed, much research conducted before the pandemic showed support for this view. A systematic review of reviews by Stiglic and Viner [14] nicely summarises this body of research. They found moderately strong evidence for positive associations between screen time and obesity (based on six reviews) and moderate evidence for an association between screen time and higher energy intake (based on three reviews) [14].

Tangential evidence looking at downstream outcomes of low physical activity further supports these findings. In a meta-analytic study involving 16 studies (with data from the North American, African, Australian, European, and Asian regions), it was found that there were significantly increased odds of obesity among adolescents who spent more than two hours on screens a day, relative to those who met the two-hour guideline [4]. In another meta-analytic study involving nine studies (four from Asia, three from South America, one from Europe, and one from North America), adolescents with greater than two hours of screen time on the weekend were found to have a greater occurrence of metabolic syndrome, although this association was not significant when the average screen time across the whole week was taken into account [33]. This highlights the possibility that not all screen time may be harmful for adolescents’ physical health.

Indeed, some have argued that certain screen-based technologies have enhanced, rather than displaced, adolescents’ physical activity. For example, online physical education (PE) lessons have been suggested to positively influence adolescents’ engagement in physical activity during the lockdown. In a study of adolescents from 10 European countries, it was found that being active in online PE lessons during the pandemic was associated with healthy levels of physical activity, especially in countries that were mildly affected (i.e., Germany, Romania, Poland, Slovenia, and Hungary) by the pandemic [15]. While the efficacy of online PE lessons has been noted to vary substantially [34], these findings in Europe provide promising evidence for the positive effects of specific types of screen time on adolescents’ wellbeing. A separate meta-analysis examining school-based eHealth interventions aimed at increasing adolescents’ physical activity supports these findings [35]. Their findings show that, at least in the short term, eHealth interventions can increase adolescents’ physical activity [35]. During the pandemic, where outdoor play is less accessible, screen-based digital technology that encourages physical activity may be particularly beneficial for adolescents.

Taken together, the pandemic has caused some shifts in the use of screen time. Although screen time was thought to displace physical activity in the past, some screen-based activities appear to encourage greater physical activity during the pandemic. It would be simplistic to argue that the implications of screen time on adolescents’ physical health today are exactly as they were pre-pandemic. To the extent that screen-based technologies may encourage greater physical activity when access to outdoor play is limited, screen time may be beneficial for adolescents’ physical wellness during the pandemic.

#### 3.5.2. Screen Time and Sleep

Another relevant outcome related to adolescents’ physical wellness is that of sleep. Specifically, much work has suggested that adolescents with higher screen time tend to have shorter sleep durations. A meta-analytic study (including samples from Asia, America, Australia, Europe, and the Middle East) examining the relationship between excessive technology use and sleep found that adolescents who use technology excessively have greater sleep problems, shorter sleep duration, and prolonged sleep onset latency [16]. Supporting these findings, a 12-country European study showed a significant association between screen time and shorter sleep duration among early adolescents, although the effect was small, whereby every extra hour of screen time was associated with 4.2 min less of sleep [36]. Data from the 2017 Youth Behaviour Survey (comprising 14,603 adolescents from the United States) similarly showed that adolescents who displayed excessive screen time behaviours were 1.34 times more likely to have less than 8 h of sleep per day [37]. 

The relationship between screen time and poorer sleep outcomes is even more pronounced when considering screen time right before bedtime. A separate meta-analysis focused on screen time before bedtime found strong and consistent associations between bedtime media device use and shorter sleep, poorer sleep quality, and greater daytime sleepiness [38], further corroborating the detrimental effects of screen time before bed. Of importance, some have even contended that the relationship between screen time and poor wellbeing outcomes is mediated by sleep [39]. For example, in a study of 2865 American adolescents, it was found that sleep mediated the relationship between screen time and depressive symptoms [39]. Theoretically, there is little reason to believe that screen time may benefit adolescents’ sleep. However, some studies have emerged that provide evidence suggesting a lack of a (practically and statistically) significant relationship between the two variables, rather than a positive relationship between screen time and sleep. Nonetheless, these results should be noted as well. Based on a large-scale study of 50,212 American children (aged 6 months to 17 years), Przybylski [40] found that every hour increase in screen time was associated with less than 10 min of change in nightly sleep duration. Considering both sides of the argument, it is without a doubt that screen time is at the very least *not* beneficial for adolescents’ sleep. In particular, it appears that screen time right before bed, relative to total daily screen time, is particularly detrimental for sleep quality and duration. 

How might the relationship between screen time and sleep be attenuated during the COVID-19 pandemic? From a theoretical perspective, we might expect that COVID-19 may have increased the prevalence of night-time digital media use, which may negatively impact adolescents’ sleep quality. Specifically, given that many day-to-day activities that occupy adolescents’ time (e.g., school lessons, homework) have shifted online in light of the pandemic [41], adolescents may be engaging in more screen-based activities throughout the day, including at night [42].

Whilst little empirical work has examined adolescents’ engagement of screen time specifically before bedtime during the pandemic, existing work suggests that the negative association between screen time and sleep quality among adolescents still holds in the COVID-19 context. For example, a cross-sectional study conducted during the COVID-19 pandemic among 243 Indonesian high school students (aged 15–18) found that greater screen time was significantly associated with a higher prevalence of sleep disorders [43]. Another study of 4314 Italian adolescents (1–18 years of age) found that adolescents had a significant delay in bedtime, as well as increased screen time (beyond that of screen time acquired during online lessons) [17]. Although it is notable that the latter study did not directly examine the association between screen time and sleep difficulty and that their findings show that adolescents’ sleep duration increased due to later risetime. 

Taken together, more research is warranted to examine if the association between screen time and poorer sleep quality has changed as a result of the pandemic. Research done outside of the North American region in particular is lacking. However, theoretically, it may be expected that the relationship between screen time and sleep quality among adolescents may have worsened post-pandemic, regardless of geographical region. It is also notable that a substantial body of literature evidences the negative influence of screen time on sleep quality, especially that of screen time right before bedtime. If adolescents’ levels of screen time engagement before bedtime increased substantially during the pandemic period, this may have negative effects on their sleep quality.

#### 3.5.3. Screen Time and Social and Emotional Wellness

Another particularly relevant outcome of screen time is the impact of screen time on adolescents’ social relationships and emotional wellbeing. In an era where physical gatherings and offline socialisation are limited, there have been debates about whether screen-based technology benefits or harms adolescents’ wellbeing [42].

On one hand, there is some evidence to suggest that social networking sites may provide users with certain forms of social support; a meta-analysis of 31 studies yielding 73 effect sizes found that using social networking sites was positively associated with emotional and informational support [44]. These positive effects were stronger among older (relative to younger), female (relative to male), and Asian (relative to Western) samples. It should be noted, however, that the studies included were not limited to those of adolescents; the average age of the samples included ranged from 15.4 to 28 years, with a majority of studies having mean sample ages above 20 years. Additionally, approximately two-thirds of the studies included were from the United States (with the other third coming from South Korea, Hong Kong, Europe, China, Taiwan, and Malaysia). Notwithstanding these limitations, this meta-analysis provides some evidence that social networking sites may be beneficial in providing some forms of social support to users.

Other preliminary findings, specifically among adolescents, further provide some support that digital media use may be beneficial for adolescents’ wellbeing. A large-scale study conducted among 910 Belgian adolescent high schoolers (average age of 15.4 years old) found that girls who actively (rather than passively) used Facebook experienced greater perceived online social support and lower depressed mood [45]. Although the same study also found that girls who use Facebook passively and boys who actively use Facebook in a public setting experience negative impacts on their wellbeing, it highlights that some groups of adolescents may benefit from social media use.

Despite these promising works, there is also strong evidence to suggest that screen time in general is associated with poorer wellbeing outcomes among adolescents. A recent systematic review examining screen time, green time, and mental health outcomes among adolescents found relatively consistent evidence that screen time was associated with increased anxiety and depressive symptoms [18]. Specifically, among studies examining early adolescents (12–14 years old), 32 of 39 studies showed at least one significant association between screen time and poorer psychological outcomes (as indicated by increased anxiety symptoms, increased depressive symptoms, greater overall mental health issues, and poorer psychological well-being, among other indicators). Similarly, among studies examining older adolescents (15–18 years old), all 13 included studies showed at least one significant association between screen time and poorer psychological outcomes. Taken together, there appears to be relatively strong evidence that adolescents’ screen time is associated with poorer mental health outcomes.

Additionally, there are concerns that the increased online social support may not be enough to replace offline social support. Highlighting this point, a study of 2528 Belgian adolescents living in the province of Antwerp showed that offline social networks were a much more important factor for subjective wellbeing, relative to online social networks [46]. Importantly, the benefits of online social networks disappeared after controlling for offline social support, suggesting that the benefits of online social networks may only arise if such online networks “bleed” into adolescents’ offline lives. 

All things considered, how might the association between screen time and wellbeing have changed in the pandemic context? For one, the existing literature appears to suggest that digital media use may increase social wellbeing among groups of adolescents that are unable to gain such support in their offline lives. During the pandemic, where physical gatherings are restricted, it appears that screen time may be beneficial in buffering the emotional impacts of physical isolation. Somewhat supporting this idea, a meta-analytic review conducted during the COVID-19 pandemic showed that one-to-one online communication (through platforms such as Skype and WhatsApp) and positive online experiences were beneficial in alleviating feelings of loneliness and stress [19]. 

Nonetheless, in the same study, the overall meta-analytic relationship between social media use and ill-being was still significant (*r* = 0.171) [19], indicating that social media use was still generally associated with poorer wellbeing. Corroborating these findings, Trott and colleagues’ [2] systematic review of the association between screen time and mental health outcomes among adolescents during the COVID-19 pandemic showed that six of nine (67%) effect sizes between screen time and anxiety were significant. Four out of four studies (100%) showed significant associations between screen time and depression. While the countries represented in the systematic reviews were unclear, and there was a jarring underrepresentation of African samples in the meta-analytic study by Marciano et al. [19], it appears that preliminary work suggests that the negative implications of screen time on emotional wellbeing remain robust in the COVID-19 context.

Taken together, some screen time appears to be beneficial for adolescents’ wellbeing in the context of COVID-19. To the extent that screen time is used for one-to-one communication and positive online experiences, screen time may allow adolescents to feel greater online social connectedness and greater wellbeing. Nonetheless, high screen time is still likely to be associated with poorer wellbeing outcomes among adolescents. As such, screen time should still be limited and regulated to maximise the benefits of digital media while minimising its negative implications on adolescents’ wellbeing.

### 3.6. Future Directions

All things considered, it is obvious that the relationship between screen time and adolescents’ holistic wellbeing is not straightforward. While screen time appears to generally be associated with poorer wellbeing and health outcomes, the purpose of using digital media plays a substantial role in influencing the relationship between screen time and wellbeing. Considering the substantial increases in screen time among adolescents during the pandemic, it is important to weigh the benefits and detrimental impacts of screen time on adolescents’ holistic wellbeing carefully. Hence, the current literature review sought to examine existing evidence regarding the association between screen time and wellbeing through synthesising highly-powered studies and systematic reviews. 

The adoption of a culturally nuanced perspective revealed invaluable insights into current research gaps as well. Much of the existing research (both pre-pandemic and during the pandemic) appears heavily reliant on adolescent samples from North America and Europe, as well as more affluent countries in Asia. Much more work is needed to examine the relationship between screen time and wellbeing among understudied populations, such as those in Africa and less affluent Asian countries. Given that technology may not be as ubiquitous in these areas, it may be possible that screen time could show greater benefits and/or less harm among adolescents from these regions. With a limited body of research, it is difficult to determine if existing findings are generalizable to these populations.

Notwithstanding these limitations in the present literature, the current work provides important insights into the relationship between screen time and wellbeing. Additionally, the present work considers how these relationships may have changed in the current pandemic context. The current literature review showed that screen time in the context of COVID-19 may:be beneficial in promoting greater physical activity (e.g., exergaming through active video games);be particularly detrimental for sleep outcomes, given possible increases in screen time before bed;be beneficial for maintaining social connectedness during lockdowns;and is still detrimental for adolescents’ wellbeing in large quantities.

Considering the complex relationship between screen time and adolescents’ holistic wellbeing, it is imperative that we move beyond the simple limits of screen time to ensure that we adopt a holistic approach to maintaining adolescents’ wellbeing. In light of the findings of the current work, we highlight that integrated guidelines, including guidelines on screen time, physical activity, and sleep, would be a particularly useful resource for parents in today’s day and age.

### 3.7. The Importance of Integrated Activity Guidelines

Integrated guidelines have been increasingly popular in recent times. The World Health Organization global guidelines on physical activity and sedentary behaviour for children and adolescents, the Canadian 24-h movement guidelines for children and youth [47], the Australian 24-h movement guidelines for children and young people, and the Singapore Integrated 24-Hour Activity Guidelines for Children and Adolescents [48] are just some examples of such guidelines. These guidelines adopt a holistic perspective towards adolescents’ wellbeing and provide guidelines on physical activity, sedentary behaviour (screen time), and sleep.

Generally speaking, children and adolescents are encouraged to have no more than 2 h of screen time, engage in at least 60 min of moderate to vigorous physical activity, and attain 9–11 h (for 5–13-year-olds) and 8–10 h (for 14–18-year-olds) of sleep daily [49]. These guidelines are informed by a substantial body of research (see [50] for a review). Ensuring all guidelines are met can help ensure that adolescents’ holistic wellbeing is taken care of. Indeed, although there is much interest and discussion about the impacts of technology on modern-day children, merely limiting screen time among adolescents is not enough for the holistic wellbeing of adolescents. As suggested by the findings of our current literature review, integrated guidelines such as those mentioned above are particularly useful in mitigating the negative impacts commonly associated with screen time but do not necessarily tackle the root of other issues (e.g., low physical activity). 

## 4. Limitations

### 4.1. Underrepresented by African and Less Affluent Asian Population

Given that the current work draws on the existing body of literature, the lack of studies examining African and less affluent Asian samples in the existing body of work carries over into the current narrative review. Indeed, our investigations show that there is a lack of highly powered studies examining the relationship between screen time and well-being among adolescents from the African and less affluent Asian regions. In contrast, most of the highly powered studies that exist in the literature examine affluent Western nations such as the United States. Thus, it is possible that the conclusions of the present narrative review may not extend to these underrepresented populations. Nonetheless, it is our hope that the present review raises awareness about the issue of these underrepresented samples so as to spur more research in this area.

### 4.2. Subjectivity of the Studies Selected for Review

Additionally, alike all narrative reviews, the current work may be subjected to bias in the studies selected for the present review. While we have made conscious efforts to ensure that the studies selected are highly powered and representative of multiple nations, as well as taken deliberate steps to search for disconfirming evidence that goes against the common conception that screen time is detrimental for adolescents’ well-being, we acknowledge that the studies included in the review may still be subjected to our unconscious biases. Thus, it is our hope that readers will still engage in their own critical thinking while reading the evidence presented in the current paper to determine if they are sufficiently convinced by the conclusions we draw from our narrative review. We agree that some readers may not arrive at the same conclusions as us, but we believe that the present review is still useful in generating greater discussion of how (and whether) the pandemic may have indeed altered the relationship between screen time and well-being among adolescents in today’s day and age.

## 5. Conclusions

In sum, the present work presents evidence to support the view that a large proportion of adolescents fail to meet the recommended screen time limit of two hours per day. Furthermore, the pandemic situation further exacerbated this situation. A close examination of the impacts of screen time on adolescent wellbeing suggests that screen time may displace physical activity, although certain digital activities may also promote physical activity in situations where physical gatherings are restricted in the pandemic. Screen time right before bedtime negatively impacts adolescent sleep, and such an impact may be worsened during lockdowns. Additionally, although certain forms of technology may improve social connectedness during the pandemic, it is important to prevent excessive screen time to mitigate the negative implications of screen time on mental health.

Parents and policymakers should also consider referring to existing integrated activity guidelines to promote a holistic sense of wellbeing among adolescents given the complex nature of screen time. Limiting screen time alone is not enough to ensure adolescents’ physical and emotional health. All in all, our hope is that the present work promotes a greater understanding of the relationship between screen time and adolescent wellbeing during and beyond the pandemic among researchers, parents, and policymakers.

## Figures and Tables

**Figure 1 sports-11-00038-f001:**
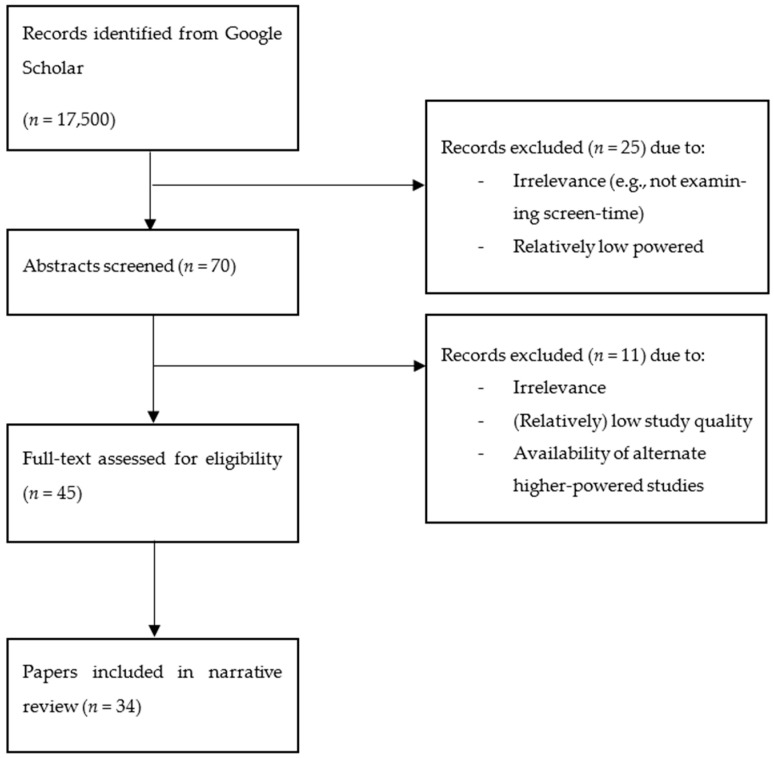
A schematic of the identification and selection of studies for the present narrative review is adapted from Ferrari [12]. The values in the figure are estimated and may not be exact. The limitations of the subjectivity of the study selected were outlined under limitations in this paper.

**Table 1 sports-11-00038-t001:** Selected studies on screen time and different domain of wellness.

	Paper	Methods	Period of Studies	Findings
Screen Time and Physical Wellness				
Stiglic and Viner (2019) [14]	Effects of screentime on the health and well-being of children and adolescents: a systematic review of reviews	Systematic review of reviews Included 13 meta-analyses examining screen time and health and well-being among children and adolescents (0–18 years old) published before February 2018.	Pre-pandemic	The six meta-analytic reviews (five of medium quality and one of low quality) examining the relationship between screen time and obesity included longitudinal studies, case-control studies, and cross-sectional studies. Altogether, there was consistent evidence that screen time was associated with greater adiposity (p. 8). The three meta-analytic reviews (two of medium quality and one of low quality) examining the relationship between screen time and obesity included experimental studies, longitudinal studies, and cross-sectional studies. Altogether, there was moderate evidence for an association between screen time (particularly television screen time) and a poorer diet (p. 10).
Kovacs et al. (2022) [15]	Physical activity, screen time and the COVID-19 school closures in Europe—An observational study in 10 countries	Large-scale observational study Included 8395 children (6–11 years old) and adolescents (12–18 years old) from Germany, Hungary, the Russian Federation, Slovenia, Spain, France, Italy, and Portugal; Parents completed the survey on behalf of children aged 12 and below.	During Pandemic	More than half of all participants were active at least sometimes during online P.E. lessons. Being active during online P.E. was associated with healthy levels of physical activity (OR = 1.27, 95% CI = [1.12, 1.44]). Subgroup analyses found that this was true for children and adolescents in mildly affected countries and adolescents from severely affected countries (See [15], Table 3, p. 1099).
**Screen Time and Sleep**				
Mei et al. (2018) [16]	Sleep problems in excessive technology use among adolescent: a systemic review and meta-analysis	Systematic review and meta-analysis Included 23 studies examining screen time and sleep among adolescents (11–20-years-old) published between 1999 and 2018; nine studies were conducted in Europe, 10 in East Asia, and four in West Asia.	Pre-pandemic	There were significant associations (*ps <* 0.05) between excessive technology use and sleep duration (standardised mean difference = −0.25, p. 3), sleep onset latency (standardised mean difference = 0.16, p. 6), and sleep problems (OR = 1.10, p. 6).
Bruni et al. (2022) [17]	Changes in sleep patterns and disturbances in children and adolescents in Italy during the COVID-19 outbreak	Large-scale cross-sectional study comparing sleep and screen time pre- and during the pandemic. Included 4314 children and adolescents (29.3% 1–3 years old, 20.7% 4–5 years old, 42.8% 6–12 years old, and 7.2% 13–18 years old) from Italy.	Pre-pandemic and during pandemic	Proportion of adolescents (6–12 years old) who had more than 2 h of screen time per day increased from 9.00% to 69.29%. Proportion of adolescents (13–18 years old) who had more than 2 h of screen time per day increased from 25.61% to 93.06% (See [17], Table 7, p. 171). Adolescents (6–18 years old) had later sleep times (See [17], Table 2, p. 169), later rise times (See [17], Table 3, p. 169), greater sleep latency (Table 5, p. 170), and greater difficulty falling asleep (See [17], Table 8, p. 172) during the pandemic compared to before the pandemic.
**Screen Time and Emotional Wellbeing**				
Oswald et al. (2020) [18]	Psychological impacts of “screen time” and “green time” for children and adolescents: A systematic scoping review	Systematic review Included 186 studies examining screen time, green time, and mental health outcomes among children (0–11 years old; *n* = 58), young adolescents (12–14 years old; *n* = 39 examining screen time), and older adolescents (15–18 years old; *n* = 13 examining screen time) published before 18 February 2019.	Pre-pandemic	Among studies examining early adolescents (12–14 years old), 32 of 39 studies found at least one significant association between screen time and poorer psychological outcomes. These outcomes included increased anxiety symptoms, increased depressive symptoms, greater overall mental health issues, and poorer psychological well-being, among others (p. 17). Among studies examining older adolescents (15–18 years old), all 13 studies found at least one significant association between screen time and poorer psychological outcomes. These outcomes included increased anxiety and depressive symptoms, among others (p. 22).
Marciano et al. (2022) [19]	Digital Media Use and Adolescents’ Mental Health During the COVID-19 Pandemic: A Systematic Review and Meta-Analysis	Systematic review and meta-analysis Included 30 studies (for a systematic review; 12 studies from Asia, 11 from Europe, three from Oceania, two from America, one from the Middle East, and one from Italy, Argentina, and the United Kingdom)/23 studies (for meta-analysis) examining the relationship between digital media use and adolescents’ (10–24-years-old) mental health during the COVID-19 pandemic before September 2021.	During pandemic	Time spent on social media was significantly positively correlated with ill-being (*k* = 11, *r* = 0.17; p. 6). The authors conclude that “social media use was helpful in mitigating the feeling of loneliness during COVID-19, but only when a one-to-one or one to-few communication (e.g., use of VoIP apps), rather than a general social media use, was promoted” (p. 24).

*Note*. These studies were selected because they were the most representative studies included in the narrative review. Specifically, these studies examined the most diverse samples and provided the most direct evidence for the relationship between screen time and wellness among adolescents.

## Data Availability

No new data was created for this study.

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
