# Peer review of "A Narrative Review of Screen Time and Wellbeing among Adolescents before and during the COVID-19 Pandemic: Implications for the Future"

_sports, 2023, doi:10.3390/sports11020038_

Round 1

Reviewer 1 Report

Thank you very much for the opportunity to review this research. 

I believe that this research needs to be considerably improved for publication. 

The introduction is very contextualised, however the motivation of the article needs to be further developed (explaining the importance of this article and how it differs from the rest).

Where were these articles sought? Why were these articles chosen? What are the reasons for discarding other research? Is there a diagram that summarises the process of exclusion and selection of articles? What methodological approach was followed?

A section entitled limitations and future perspectives needs to be added. 

Regarding the conclusions all methodological flaws make the conclusions drawn unreliable. 

Reviewer 2 Report

First, I would like to recognise the authors for the review they generated on the factors by which screen-time affects well-being among adolescents, and how the pandemic may have influenced some of these factors.

The title is clear and informative

The abstract is very well written

Line 6 of the abstract: you wrote mfactors. I think you probably mean factors.

Line 9-10 of the abstract: ‘It is important to ensure that beyond ensuring that adolescents limit…’. Consider rephrasing this sentence or changing ensure to affirm… this is just a suggestion, and it is preferable to use your own words but to the reader, it is a little confusing to follow the ‘ensure that beyond ensuring’

The introductory section is fine. The methodology section is describing the inclusion criteria for the studies but fails to present where the studies were taken from. Search and eligibility criteria were not mentioned, such as which databases did you use? (PubMed and web of science?). Which search terms did you use to find the included studies?

A flow chart diagram for the study selection and data extraction would have been nice to present. This is minor, however. As long as you explain where and how you got the studies it will be fine.

Round 2

Reviewer 1 Report

The authors have significantly and considerably improved the research. They have carried out a great improvement of the research compared to how it was in the first stage. Therefore, taking into account all of the above, I consider this study to be publishable. 

Author Response

We thank the reviewer for the positive evaluation, and for the constructive comments that have helped to substantially improve the manuscript.